# CryoEM structure of the Nipah virus nucleocapsid assembly

**De-Sheng Ker** [id], **Huw T. Jenkins** [id], **Sandra J. Greive**[¤], **Alfred A. Antson** [id] *

York Structural Biology Laboratory, Department of Chemistry, University of York, York, United Kingdom

¤ Current address: Dreampore SAS, 33 Boulevard du Port, Cergy-Pontoise, France
* fred.antson@york.ac.uk

**Data Availability Statement:** The coordinates have been deposited with the Protein Data Bank under accession codes 7NT5 (helical nucleocapsid) and 7NT6 (clam-shaped assembly); the corresponding cryoEM reconstructions have been deposited with

## Abstract

Nipah and its close relative Hendra are highly pathogenic zoonotic viruses, storing their ssRNA genome in a helical nucleocapsid assembly formed by the N protein, a major viral immunogen. Here, we report the first cryoEM structure for a *Henipavirus* RNA-bound nucleocapsid assembly, at 3.5 Å resolution. The helical assembly is stabilised by previously undefined N- and C-terminal segments, contributing to subunit-subunit interactions. RNA is wrapped around the nucleocapsid protein assembly with a periodicity of six nucleotides per protomer, in the "3-bases-in, 3-bases-out" conformation, with protein plasticity enabling non-sequence specific interactions. The structure reveals commonalities in RNA binding pockets and in the conformation of bound RNA, not only with members of the *Paramyxoviridae* family, but also with the evolutionarily distant *Filoviridae* Ebola virus. Significant structural differences with other *Paramyxoviridae* members are also observed, particularly in the position and length of the exposed α-helix, residues 123–139, which may serve as a valuable epitope for surveillance and diagnostics.

## Author summary

Nipah virus is a highly pathogenic RNA virus which, along with the closely related Hendra virus, emerged relatively recently. Due to ~40% mortality rate and evidence of animal-to-human as well as human-to-human transmission, development of antivirals against the Nipah and henipaviral disease is particularly urgent. In common with other single-stranded RNA viruses, including Ebola and coronaviruses, the nucleocapsid assembly of the Nipah virus safeguards the viral genome, protecting it from degradation and facilitating its encapsidation and storage inside the virion. Here, we used cryo-electron microscopy to determine accurate three-dimensional structure for several different assemblies of the Nipah virus nucleocapsid protein, in particular a detailed structure for the complex of this protein with RNA. This structural information is important for understanding detailed molecular interactions driving and stabilizing the nucleocapsid assembly formation that are of fundamental importance for understanding similar processes in a large group of ssRNA viruses. Apart from highlighting structural similarities and differences with nucleocapsid proteins of other viruses of the *Paramyxoviridae* family, these data will inform the development of new antiviral approaches for the henipaviruses.

the Electron Microscopy Data Bank under accession codes EMD-12581 and EMD-12584.

**Funding:** This work was supported by the Wellcome Trust award 206377 to AAA. DSK was supported by Platinum Scholarship from Tony Wild fund, www.york.ac.uk/chemistry/postgraduate/research/funding/wild-fund-scholarships/. CryoEM data collection for this work was performed at the UK national electron bio-imaging centre (eBIC), proposal EM19832-12, funded by the Wellcome Trust, MRC and BBSRC. The funders had no role in study design, data collection and analysis, decision to publish, or preparation of the manuscript.

**Competing interests:** The authors have declared that no competing interests exist.

## Introduction

Nipah virus (NiV) and the closely related Hendra virus are emerging zoonotic RNA viruses that cause a range of illnesses, from asymptomatic infections to atypical pneumonia and fatal encephalitis. NiV was first discovered during an outbreak of encephalitis among pig farmers in peninsular Malaysia in 1998 with a case-fatality rate of ~40%[1]. Whilst the *Pteropus* fruit bats serve as the natural reservoir host for NiV, the virus was shown to infect a wide range of animals, with evidence of human to human transmission[2–4]. As one of the most serious emerging infectious diseases, NiV disease has been included among a shortlist of the blueprint priority diseases by the World Health Organisation (WHO), alongside Ebola virus disease, Zika and severe acute respiratory syndrome (SARS). However, currently there is no approved treatment or vaccine available.

Along with the Hendra virus, which was discovered in Australia in 1994, NiV is a member of the newly delineated *Henipavirus* genus in the *Paramyxoviridae* family. As with other paramyxoviruses, the NiV RNA genome conforms to the "rule of six", where the length of the genome is a multiple of six nucleotides[5]. The NiV RNA genome is encapsidated by the nucleoprotein (N), forming a long helical nucleocapsid assembly[1]. This assembly not only safeguards the viral genome from degradation, but also serves as a template for productive transcription of mRNA and replication of the nascent viral RNA genome by the viral RNA dependent RNA polymerase (RdRp)[6]. During the early stage of the *Paramyxoviridae* viral replication cycle, viral mRNA transcription predominates and the viral mRNA is translated into the viral proteins by the host translation machinery. After sufficient amounts of the N proteins have been produced, viral genome replication occurs[7]. The newly synthesized viral RNA genome is encapsidated co-transcriptionally by nascently translated N protein, driving the formation of a helical nucleocapsid assembly, making the N protein one of the most abundant viral proteins produced during infection. During virion formation, the nucleocapsid and RdRp complex are transported to the plasma membrane for assembly and budding in a process that is mainly driven and coordinated by the interaction of the viral matrix protein (M) with the glycoproteins and the nucleocapsid assembly[8,9]. RNA binding by the N protein to form the nucleocapsid, therefore, is an essential step in viral assembly and understanding this process would inform the development of antivirals. The N protein is also a highly immunogenic antigen, partly due to its high abundance during infection, making it a vital tool for the serological surveillance required for diagnostic and epidemiological studies of new and historic outbreaks[10,11].

Structures of nucleocapsid-like assemblies of several paramyxoviruses, assembled as helical [12], ring[13] or clam-shaped[14] complexes, have been already reported. However, the nucleocapsid of NiV shares only 32% sequence identity with the nucleocapsid of the Measles virus, the closest homologue with an available structure. Although the overall fold would be expected to be preserved, the extent of structural differences due to this considerable evolutionarily divergence would make it challenging to precisely locate antigenic regions, and relate biochemical findings to the structure, using only a homologous structure. The only structural information for the NiV nucleocapsid is available for a truncated single subunit lacking N-terminal (residues 1–31) and C-terminal (residues 384–532) regions, which was determined in complex with a 50 amino acid peptide of the P protein[15]. No structural information is available for the oligomeric assembly of the NiV nucleocapsid, despite the ability of the recombinant protein expressed in bacteria[16], yeast[17] and insect cells[18] to form nucleocapsid-like helical structures containing cellular RNA. Here, we report CryoEM structures of several different types of assemblies formed by recombinantly produced NiV N protein, elucidating detailed information about protein-RNA interactions. These structures also reveal how the N- and C-terminal segments of the NiV N protein, which were not present in the construct used

to determine the previous N protein structure, stabilise the assembly, by interacting with the same surfaces of adjacent subunits that were previously shown to interact with the P-protein [15]. These new data permit the analysis of similarities and differences with other members of the *Paramyxoviridae* family as well as more distantly related members from the *Mononegavirales* order of ssRNA viruses.

## Material and methods

### Expression and purification of the NiV nucleocapsid spiral assembly

The NiV nucleocapsid gene, a kind gift from Wen Siang Tan at the Universiti Putra Malaysia, was cloned into the pET-YSBL-Lic expression vector. N protein was expressed (at 16˚C) in *E. coli* BL21 Gold (DE3) Rosetta pLysS grown in 2 L of LB media. Cell pellets were resuspended in 50 mL of lysis buffer containing 20 mM Tris-HCl, pH 8.0, 1 M NaCl, 1 M Urea, 50 mM Imidazole, and 10% (v/v) glycerol. Cells were lysed by sonication and the lysate clarified by centrifugation at 25,000 *g* for 30 min. The supernatant was then applied to a 5mL HisTrap FF column (GE Healthcare) which had been equilibrated with 5 column volumes (CV) of binding buffer containing 20 mM Tris-HCl, pH 8.0, 1 M NaCl, 50 mM Imidazole, and 10% (v/v) glycerol. The column was washed with 10 CV of binding buffer containing 50 mM Imidazole, followed by 6 CV of binding buffer containing 100 mM Imidazole. The protein was eluted using a linear gradient from 100 mM to 500 mM imidazole over 20 CV. Eluted proteins was concentrated and further purified by size exclusion chromatography (Superose 6, GE Healthcare) in 20 mM Tris pH 8.0, 500 mM NaCl. The protein was concentrated to 1 mg/mL, flash frozen in liquid nitrogen, and stored at -80˚C. The concentration of the N protein was determined using the Bradford Assay (Thermo Fisher Scientific).

### Negative stain EM

3 μL of sample was applied to glow-discharged continuous carbon grids and stained with 2% (w/v) uranyl acetate. Negative stained grids were imaged on a Tecnai 12 BioTWIN G2 transmission electron microscope (FEI) operating at 120 keV and equipped with SIS Megaview III CCD camera. Images were recorded at a magnification of x49,000 and a defocus set to -1 μm.

### CryoEM sample preparation and data acquisition

The purified N protein was prepared on UltraAuFoil R1.2/1.3 gold support grids (Quantifoil). 3 μl of sample was applied to glow-discharged grids, blotted for 2 s with -10 force, and vitrified by plunging into liquid ethane using the FEI Vitrobot Mark IV at 4˚C and 100% relative humidity. Micrographs were collected at the Diamond eBIC facility on a Titan Krios microscope (FEI) operating at 300 keV and equipped with K2 camera and an energy filter slit width of 20 eV (Gatan). Automated data collection was performed using FEI EPU software. 1879 movies with a total electron dose of ~41 e-/Å$^2$ were recorded in counting mode over 11 s (40 frames) with a pixel size of 1.048 Å. The defocus range chosen for automatic collection was 0.5 to 2.1 μm.

### Image processing

All datasets were processed in RELION 3.0[19] unless stated otherwise. Micrographs were first motion-corrected using MotionCor2[20]. CTF parameters were estimated using CTFFIND4 [21]. Autopicking was performed in RELION using references generated from manually picked particles. All the micrographs were manually inspected to ensure picking of rare views. A total of 217,522 particles were extracted and subjected to reference-free 2D classification to remove particles associated with noisy or contaminated classes. The resulting 189,662 particles

were subjected to 3D classification using a map generated from the Measles N protein (EMDB:0141)[22], trimmed to a single turn helix using UCSF Chimera's "Volume Eraser" function[23] and low-pass filtered to 60 Å, as a reference model. The best 3D class was low-pass filtered to 30 Å and used as a reference model for a new round of 3D classification against the same initial set of particles. Reconstruction of the final spiral map was achieved by using all particles from the spiral classes (124,891 particles) with a 13-protomer spiral turn solvent mask and imposing local symmetry. This improved the map quality enabling model building. For the local symmetry, masks around all 13 protomers were created and low-pass filtered to 15 Å (S2A Fig). The local symmetry operators were generated from the search feature of the *relion_local_- symmetry* and were applied during the 3D refinement using a regularization T-value of 13 in RELION. Subsequent per-particle CTF refinement and Bayesian polishing in RELION 3.1beta led to a final map of 3.5 Å resolution, estimated by the 0.143 FSC criterion (S2B Fig). The maps were postprocessed in RELION 3.1beta[24] and are shown after B-factor sharpening.

The remaining 64,771 non-spiral particles were further subjected to 3D classification using the "clam-shaped" 3D class, obtained from previous 3D classification, as a reference model. Two major 3D classes, a spiral clam-shaped assembly (23,029 particles) and a semi-spiral clam assembly (18,979 particles), were selected. Subsequent per-particle CTF refinement in RELION 3.1beta, 3D refinement of these two 3D classes resulted in final maps of 4.3 Å (spiral clam-shaped assembly) and 5.2 Å (semi-spiral clam-shaped assembly) resolution, respectively. The local resolution maps were calculated using RELION 3.1beta.

## Model building, refinement and analysis

Atomic model building of the NiV nucleocapsid spiral assembly was performed using the previously reported crystal structure of the RNA free NiV nucleocapsid (pdb:4co6)[15] as an initial model, which was docked as a rigid body into the 3.5 Å resolution CryoEM maps using UCSF Chimera's "Fit in map" function[23]. The RNA chain was modelled as poly-uridine, and the model was adjusted manually using Coot[25]. Model refinement was performed using REFMAC5[26], phenix.real_space_refine[27], ISOLDE[28] and ERRASER2[29] (https://new. rosettacommons.org/docs/latest/ERRASER2). Terminal residue segments 1–4 and 399–532 were not modelled owing to a lack of interpretable map features.

For the "spiral clam" CryoEM maps, the monomeric model of the RNA-bound NiV N protein (taken from this study) was fitted as a rigid body into the maps[23]. No protein models were fitted into the seam regions, due to the lack of interpretable map features in the CryoEM maps. Both models were refined using REFMAC5[26] and phenix.real_space_refine[27].

Protein interfaces were analysed using the COCOMaps server[30]. Protein domain motion was analysed using the DynDom server[31]. Multiple sequence alignments were performed using Clustal Omega[32] and visualised in JalView[33] and ESPript 3.0[34]. Phylogenetic tree was conducted in MEGA X[35]. The sequence identity matrix was performed using MatGAT [36]. The conserved regions in the structure were analysed using the ConSurf server[37]. Figures showing protein/RNA structure were created using UCSF ChimeraX[38]. To calculate the electrostatic potential, the PDB format files were converted to PQR format with the PDB2PQR server using the PARSE force field and assigned protonation states at pH 7.0. The file was applied to the APBS server by including 0.15 M of ions in the calculation[39].

## Results

### Structure of the helical assembly

A spiral assembly of the full length NiV nucleocapsid protein, bound to *E. coli* cellular RNA, was purified from recombinant expression in *E. coli*, and its structure was determined by

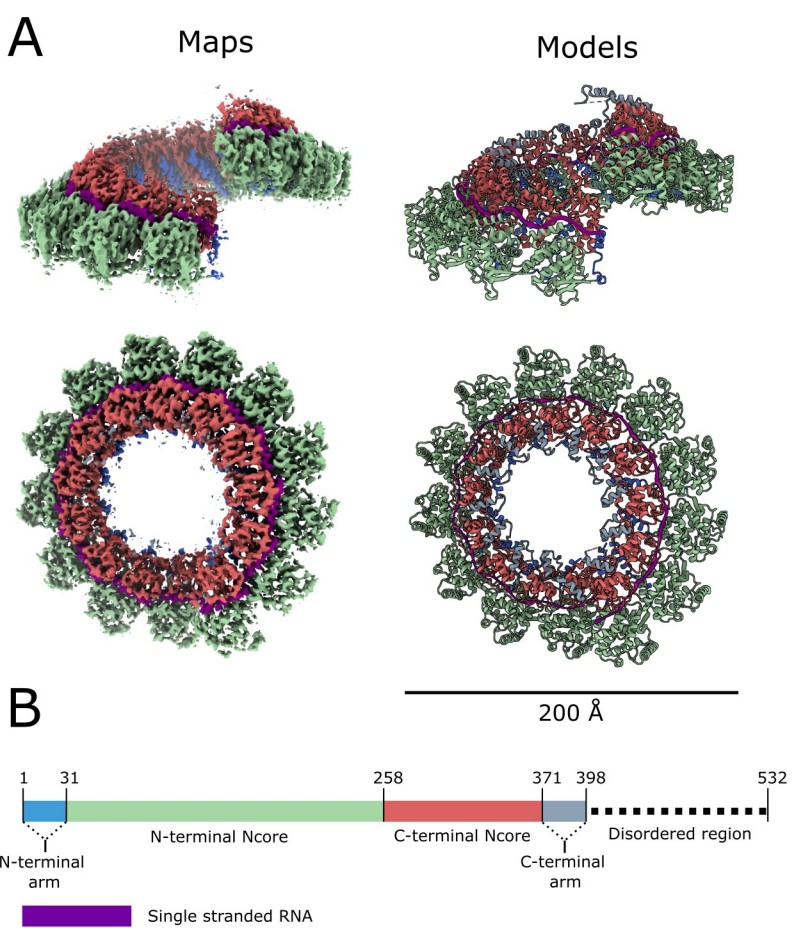

**Fig 1. Structure of the NiV nucleocapsid protein-RNA complex.** CryoEM structure of the nucleocapsid complex, determined at 3.5 Å resolution. Two orthogonal views of the CryoEM map (left) are shown next to the corresponding molecular models (right). (B) Schematic representation of the nucleocapsid protein, with the colour-coding as in (A). Boxed segments correspond to regions with defined structure. Dashed line corresponds to the disordered region.

CryoEM single-particle 3D reconstruction. 2D and 3D classification showed that the majority of the particles represent a spiral assembly comprised of 13 subunits per turn, with minor populations of particles representing a longer spiral with multiple turns, and a clam-shaped, face-to-face assembly of two short spirals[14] (S1 Fig). Reconstruction of the spiral assembly (65% of the particles) with a mask corresponding to a 13-subunit spiral turn and imposing local symmetry, resulted in a 3.5 Å resolution CryoEM map (S2 Fig). Angular distribution analysis demonstrated that, while there was a preferred orientation for the particles (viewed along the central axis of the helical assembly), there was a good distribution of particles across other orientations, including side-views (S2C Fig). The final model shows that 13 nucleocapsid monomers bind to the single-stranded RNA forming a left-handed spiral turn with outer and inner diameters of 204 and 56 Å, respectively (Fig 1). Assuming the NiV nucleocapsid forms a continuous spiral with the same symmetry, the pitch for NiV nucleocapsid is calculated to be 54 Å with 13.4 subunits per turn (S3 Fig). Each N protein monomer is comprised of two main globular N-terminal and C-terminal Ncore (Nucleocapsid core) domains, with each domain flanked by the N-terminal arm (NT-arm, residues 1–31) and the C-terminal arm (CT-arm, residues 373–398) subdomains (Fig 1B). The CT-arm is followed by a disordered C-terminal region (residues 399–532)[40] for which there is no clearly defined density.

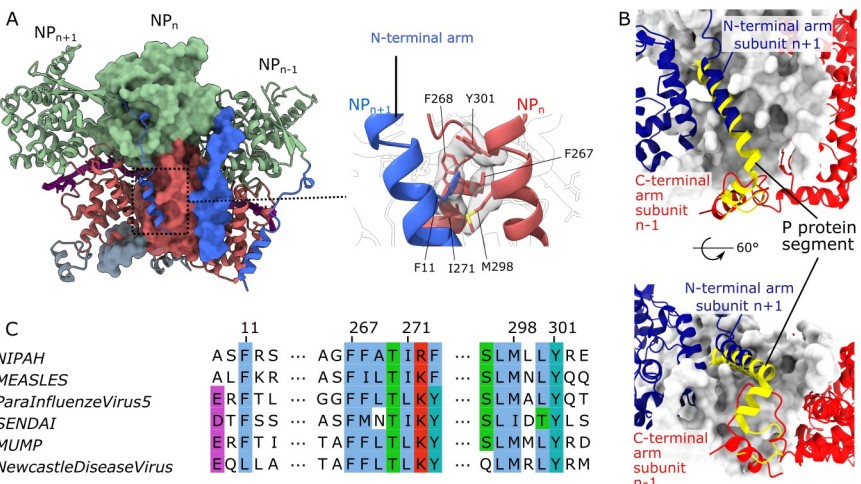

**Fig 2. Protomer-protomer interactions within the NiV nucleocapsid assembly.** (A) Three adjacent protomers, where the two outside protomers are presented as ribbons and the central protomer is shown in surface representation, with the NT-arm in blue, N-terminal Ncore in green, C-terminal Ncore in coral and CT-arm in grey, as in Fig 1B. A magnified view of the molecular interaction of the NT-arm and the C-terminal Ncore domain is shown on the right with interacting residues (sticks) displayed within the CryoEM map. (B) The P protein segment from the structure of the complex with the monomeric NiV N form (yellow ribbon, pdb:4co6)[15] superimposed onto the central subunit in (A), shown in white surface representation. The two adjacent protomers are presented as dark-blue (n+1) and red (n-1) ribbons. (C) Alignment of interacting residue segments, shown in the magnified view in (A), for N proteins from several Paramyxoviruses, with conserved residues highlighted using the ClustalX colour scheme.

## Protomer-protomer interactions within the helical assembly

The spiral assembly of the NiV nucleocapsid is primarily formed through lateral contacts over a calculated interface area of ~3000 Å$^2$ between two adjacent protomers (S1 Table). The contact area includes a hydrophobic core comprising an aromatic residue (F11) from one protomer and a triad of aromatic residues (F267, F268, Y301) in the adjacent protomer (Fig 2A). All of these aromatic residues are well conserved in the *Paramyxoviridae* facilitating similar protomer-protomer interactions across all family members[12]. The NT and CT arms, which have been reported to play a role in the spiral assembly of nucleocapsid[15,41], occupy a hydrophobic groove in the C-terminal Ncore domain of the adjacent protomer. In the crystal structure of the NiV N protein monomer (pdb:4co6), a similar hydrophobic groove is occupied by a 50 amino acid segment of the NiV phosphoprotein (P) which is essential in maintaining the N protein in its RNA-free, monomeric state (Fig 2B)[15]. Structures of the N-terminal (residues 4–31) and C-terminal (residues 372–398) segments were not resolved in the earlier study of the monomeric RNA-free form in complex with the P-protein segment, which was based on a protein construct comprising residues 32–383[15]. Both of these segments, missing in the earlier study, are critical for the oligomer assembly, making bridging interactions with adjacent subunits (Fig 2B).

## Protein-RNA interactions

The CryoEM map shows clear density for the single stranded RNA, modelled as a poly-uridine chain, wrapped around the nucleocapsid. The RNA molecule is bound to the protein in the classical "3-base-in, 3-base-out" conformation[12], where the RNA chain twists about 180° every 3 nucleotides to place three consecutive nucleotides with the bases facing the protein, followed by 3 nucleotides with exposed bases. The structure shows that the nucleic acid lies within the charged cleft of the N protein at the interface between the N-terminal Ncore and

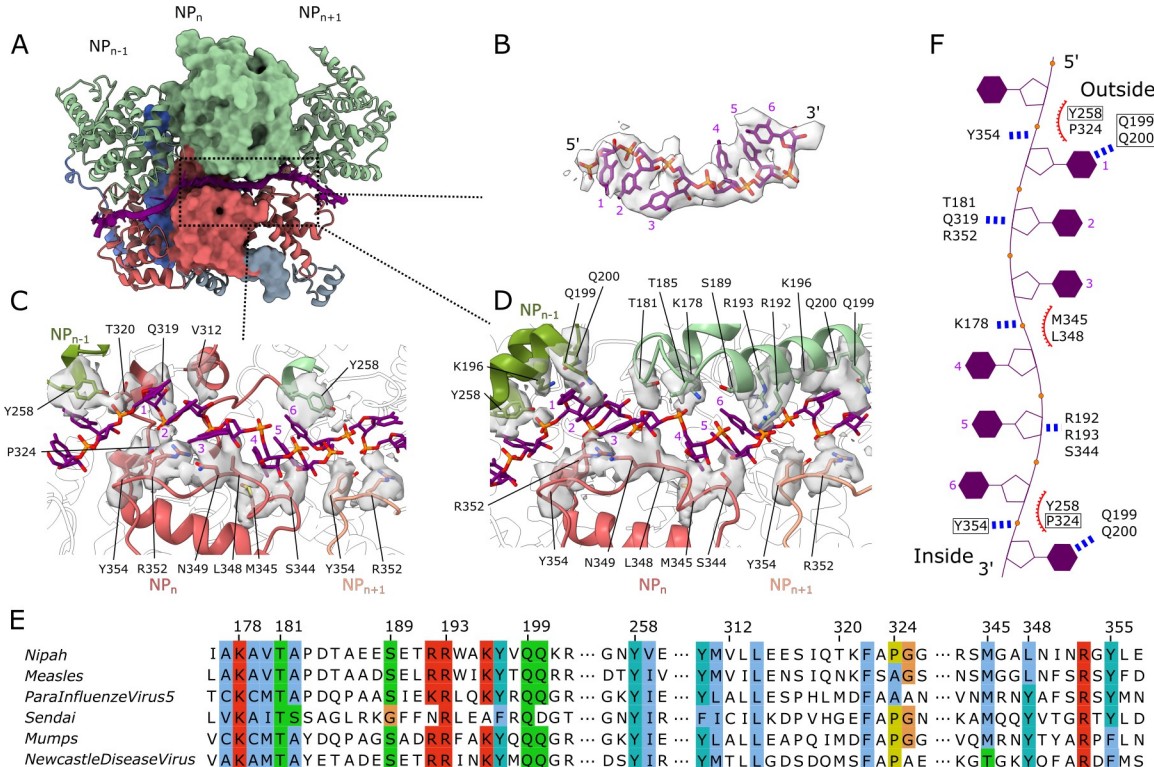

**Fig 3. Protein-RNA interactions.** (A) Three adjacent protomers shown as in Fig 2A, with RNA (purple) shown in ribbon and sticks. (B) CryoEM map corresponding to the RNA, with a fitted poly-uridine RNA model (sticks). (C-D) Two different views at the protein-RNA interface, detailing protein-RNA interactions. CryoEM density corresponding to the side chain atoms of interacting residues, is shown with a 3 Å distance cut-off. (E) Alignment of RNA-binding residue segments from several Paramyxoviral N proteins with conserved residues highlighted using the ClustalX colour scheme. (F) Schematic of the ssRNA conformation in complex with the N protein. The residues in close proximity to RNA are labelled. Boxed residues indicate those from the neighbouring protomer; the blue dotted lines indicate putative hydrogen bonding interactions; red curves indicate putative hydrophobic interactions.

the C-terminal Ncore domains. This groove is lined by the residue segments K178-Q200 and S344-Y354 that are positioned at the outer edge of the spiral assembly (Fig 3). Within the RNA binding cleft, a series of basic (K178, R192, R193, R352) and polar (T181, Q319, S344) residues, with well-defined density, are within hydrogen-bonding distance from the RNA sugar-phosphate backbone. Residues Q199 and Q200 from helix H8 also have well-defined density, with their side chains projected toward the RNA bases. These two amino acids are conserved in the nucleocapsid proteins of the measles virus (MeV) and parainfluenza virus 5 (PIV5), where they make similar interactions with RNA bases [13,22]. At the interface between the two protomers, aromatic residues Y258 and Y354, one from each adjacent protomer, are positioned in close proximity to the RNA chain (Fig 3C), facilitating the twist in its conformation. This twist in the sugar-phosphate backbone is assisted by a series of additional protein-RNA interactions contributed by several other polar residues lining the RNA binding cleft. The second twist in the RNA conformation, spaced by three nucleotides from the first one, occurs in a cleft within a single protomer and is facilitated by steric hindrance from the side chain of L348 (Fig 3D). The majority of residues interacting with the RNA within the RNA-binding cleft are highly conserved among Paramyxovirus N proteins (Fig 3E) indicating a similar mechanism for RNA coordination.

Several residues connecting the two RNA binding segments (K178-Q200 and S344-Y354) with the rest of the N-protein are poorly defined. It is likely that the flexible nature of these

regions serves to provide plasticity to accommodate and interact, in a non-sequence specific manner, with the varying sequence along the entire length of the RNA strand. This flexibility may also allow the RdRp to access the ssRNA while bound within the nucleocapsid assembly, for RNA synthesis.

In the crystal structure of the RNA-free monomeric NiV N protein, the flexible loop, residues A180-R192, was mostly disordered and positioned such that it would block access to the RNA binding cleft, suggesting that this loop needs to move out of the cleft to permit RNA binding. As seen from structure comparison, RNA binding is also accompanied by an approximately 28° rotation of the N-terminal and C-terminal Ncore domains towards each other, around a hinge region formed by the H12-H13 loop[15], H15-H16 loop and helix H17 (S4 Fig and S2 Table). Similar conformational changes have also been observed for the nucleocapsid of MeV[12].

## Comparison with the structure of N protein from other Paramyxoviruses

Within the Paramyxovirus family, the NiV N protein shares about 56%-92% sequence identity with other members of the *Henipavirus* genus, with the N proteins from Hendra virus (92% sequence identity) and Cedar virus (59% sequence identity) being the closest relatives (S5 Fig and S3 Table). Mapping these sequence differences to the structure (Fig 4) shows that the RNA-binding surfaces are the most conserved, not only among the henipaviruses, but also for nucleocapsids of more distantly related viruses such as Measles (S6 Fig). In contrast, the most variable region is the outer wing area of the assembly, constituted by the N-terminal Ncore domains of each subunit, where the most exposed part is largely made up of the 51-amino acid segment 106–156. This segment has a well-defined structure, but its conformation displays significant variability among the paramyxovirus family members, with considerable differences in the position and length of helix H5, residues 123–139 in NiV N (Fig 4, inset). Within the outer wing, the closely related NiV and Hendra virus N proteins contain four amino acid substitutions: V70I, V108L, D137E and I236M (S6 Fig), with V108 and D137 being exposed at the outermost edge.

## Clam-shaped assemblies of recombinant NiV N protein

Aside from the common helical assembly, about 35% of N protein particles were found as clam-shaped assemblies which can be further subclassified into two distinct primary conformations, a spiral clam-shaped assembly, and a semi-spiral clam-shaped assembly. The spiral clam-shaped assembly is composed of two N protein spirals stacked face to face, as seen for the Newcastle Diseases virus (NDV) N protein assembly[14]. In contrast, the semi-spiral clam-shaped assembly features one 14-subunit ring and one 13-subunit N protein spiral stacked as shown on S7 Fig. Asymmetric reconstruction of both assemblies leads to 4.3 Å and 5.2 Å resolution maps, respectively. Models for the spiral assemblies were built by rigid-body fitting and real space refinement of the N protein protomer taken from the protein-RNA complex described above. The structure of the N protein monomer within these clam-shaped assemblies remains largely the same as in the spiral assembly, with an overall RMSD of 0.9 Å calculated over Cα atoms.

For both assemblies, there is a significant surface area buried at the interface between the two halves of the clam shell, with up to ~670 Å$^2$ of buried area per monomer (Fig 5A). Interactions across this interface are mediated by hydrogen bonding and polar interactions made by loop segments A1-H2, A2-H5 and H6-H7 (Fig 5B) from each opposing protomer. The surface area, buried at the clam-shell interface of each monomer, is about five times larger than seen in the clam-shaped assembly of NDV, where only one protein loop (residues 104–124) is involved

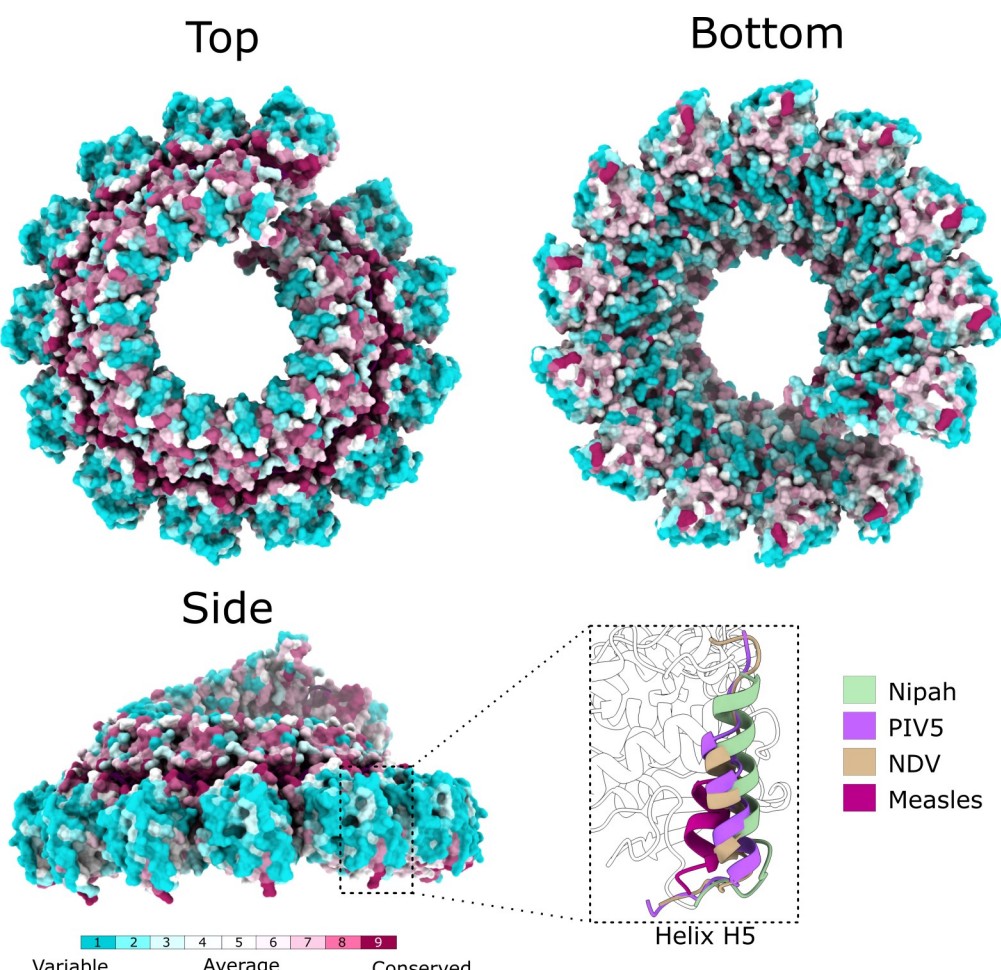

**Fig 4. Mapping sequence variation to the structure.** The surface of the NiV nucleocapsid assembly is coloured according to sequence conservation among Paramyxoviral N proteins based on ConSurf analysis, visualised at three different orientations. Inset (bottom right) compares the conformation of the residue segment 118–139 containing helix H5, in four different Paramyxoviruses: NiV, Parainfluenza virus 5 (PIV5, PDB code 4xjn), Newcastle Disease virus (NDV, PDB code 6jc3) and Measles (PDB code 6h5q).

in the interaction[14]. This likely creates a closer interaction between the halves of the NiV clam shell as compared to the NDV assembly, similar to that observed in the recently reported clam-shaped assembly of the Sendai virus (SeV)[42]. Interestingly, although sequences of these clam-shell interface loops are not conserved between NiV, NDV and SeV, or other members of the *Paramyxoviridae*, these loops are rich in glycine and proline which can facilitate conformational flexibility (Fig 5C).

## Discussion

Formation of a helical nucleocapsid assembly, that safeguards the viral genome and serves as a template for RNA replication, is a unifying feature of negative strand RNA viruses. We determined the CryoEM structure for the assembly of full-length recombinant NiV N protein with *E. coli* cellular RNA, presenting the first high resolution structural data on the nucleocapsid assembly for this virus and the *Henipavirus* genus. The RNA strand, accommodated in the groove between the N- and C-terminal Ncore domains, binds with the bases of consecutive

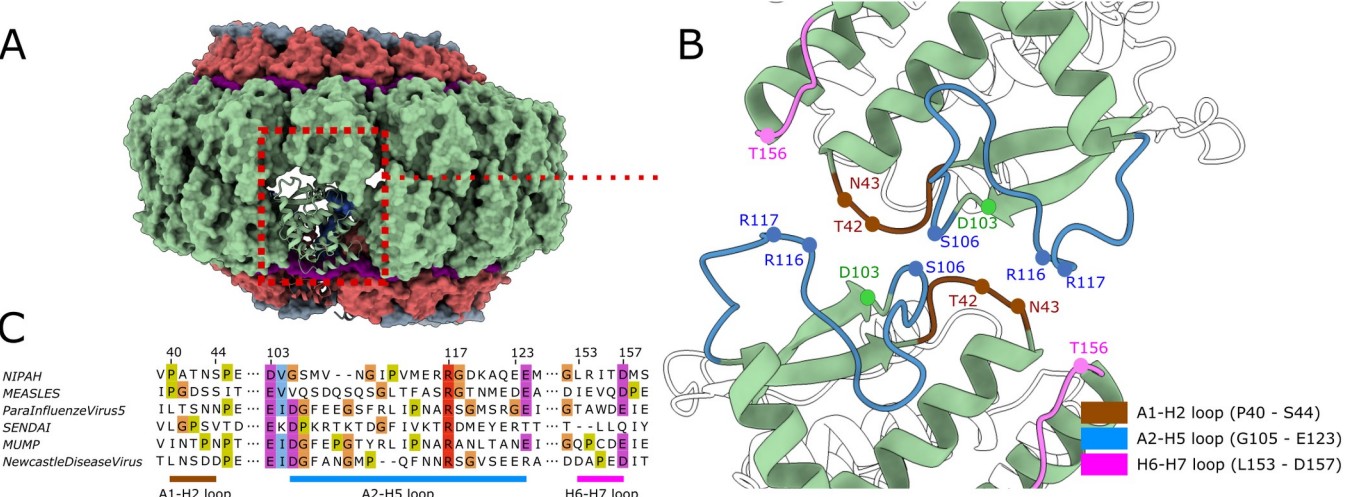

**Fig 5. Clam-shaped nucleocapsid assembly.** (A) Model of the clam-shaped nucleocapsid assembly, presented as a molecular surface with subunits coloured as in Fig 2A, showing the interaction between two opposing, top and bottom N protein spirals, with one of the protomers shown as ribbon. (B) The clam-shaped interaction is primarily mediated by protein loops from the N-terminal Ncore domain. Putative residues involved in the interaction between the two halves of the shell are indicated. (C) Alignment of the three interacting loop regions highlighted in (B) for N proteins from several Paramyxoviruses, with glycine, proline and surrounding conserved residues highlighted using the ClustalX colour scheme.

nucleotide triplets alternatively exposed and buried (Fig 3). The spiral assembly is stabilised, not only by the protein-RNA interactions, but also by subunit-subunit interactions between contiguously bound N proteins (Fig 2).

The helical assembly has 13.4 subunits per turn, in common with Measles, Parainfluenza virus 5, and Newcastle disease virus, three of the distant homologues of *Paramyxoviridae* with available structures of nucleocapsid assemblies[12–14] with which NiV N protein shares 32%, 28% and 29% sequence identity (S3 Table), respectively. Areas with the highest sequence conservation are found at the RNA-binding surfaces and also at subunit-subunit interfaces, with the outer exposed surfaces of the nucleocapsid displaying the most sequence variation (Fig 4).

The oligomeric assembly, protein-RNA interactions and the conformation of bound RNA are conserved among the different genera of *Paramyxoviridae* (Fig 6). Moreover, the N protein of Ebola virus, a representative of *Filoviridae*, binds RNA in a conserved manner, wrapping it around the outer edge of the oligomeric N protein helix, with similarities observed in protein-RNA interactions and even in the conformation of the bound RNA[43,44] (Fig 6). N proteins of other *Mononegavirales*, from families that are more distant than *Filoviridae*, also display a highly conserved protein fold, with the highest similarity observed in the N-terminal domain and more limited, but still detectable, fold similarity within the C-terminal domain[45]. Available structural information indicates that although N proteins of all *Mononegavirales* bind RNA within a groove between adjacent domains, a significant variation is observed in the number of RNA nucleotides bound per protein protomer (6 to 9 nucleotides) and also in the relative arrangement of subunits, where the number of subunits per helical turn can vary from 10 to 24, depending on the virus. Interestingly, crystal structures of circular assemblies of the rabies virus and vesicular stomatitis virus [46,47], members of *Rhabdoviridae* family, indicated that the RNA-binding groove faces towards the inside of the central tunnel. However, within the helical assembly in virions [48,49] the individual N proteins are tilted, compared to their organization in the rings and the RNA is not positioned inside the helicoidal assembly, although it does bind closer to the central axis than in the viruses of the *Paramyxoviridae* (e.g. NiV), *Filoviridae*[43,50] and *Pneumoviridae*[51,52] families, where the RNA-binding grove is

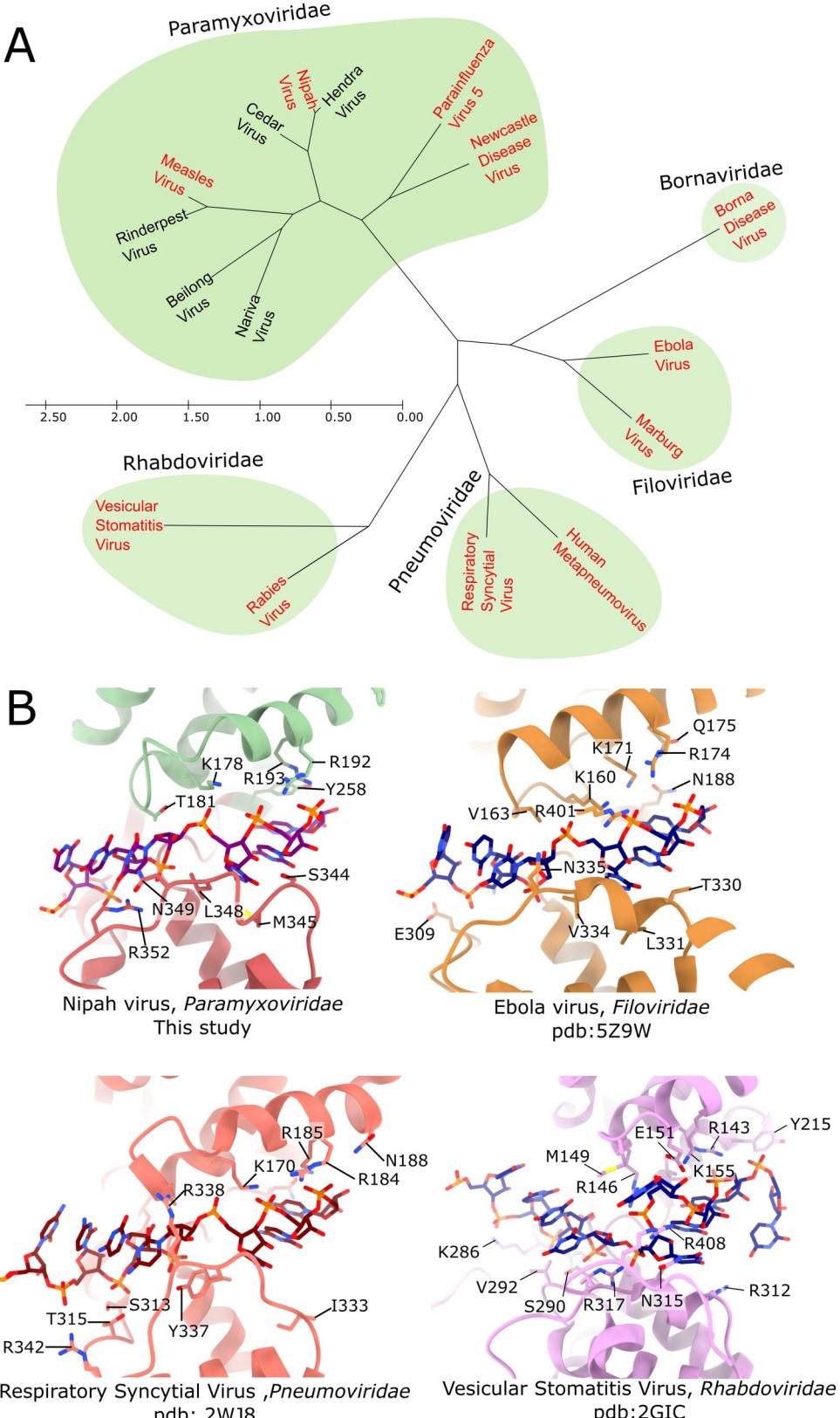

Nipah virus, *Paramyxoviridae*
This study

Ebola virus, *Filoviridae*
pdb:5Z9W

Respiratory Syncytial Virus ,*Pneumoviridae*
pdb: 2WJ8

Vesicular Stomatitis Virus, *Rhabdoviridae*
pdb:2GIC

**Fig 6. Phylogenetic relation of the N protein within the Mononegavirales.** (A) Phylogenetic tree based on the N protein sequences of selected Mononegavirales. N proteins with available protein structures are coloured in red. (B) RNA binding region of Nipah virus, Ebola virus, Respiratory Syncytial Virus, and Vesicular Stomatitis Virus showing the conserved protein-RNA interaction among these viruses. Residues involved in protein-RNA interactions are shown as sticks.

located closer to the outer edge of the helical assembly. It is interesting to note, that compared to *Paramyxoviridae* and *Pneumoviridae*, *Rhabdoviridae* also adopt a different strategy to inhibit non-specific RNA binding of its N protein, by inserting the P protein directly into the RNA binding groove [15,53,54].

Apart from significant differences between the nucleocapsid assemblies of different members of the *Mononegavirales*, structural differences are observed also for individual viruses. In particular, for paramyxoviruses, variations have been observed in the helical pitch and diameter of the assembly, depending on the pH and salt concentration[55], as well as, the presence of the C-terminal disordered region[56]. Likewise, the recombinant full length NiV N protein helical assembly is highly flexible, with further 3D classification showing significant conformational and compositional heterogeneity in the NiV N protein spiral assembly within this CryoEM dataset (S1 Fig). While most of the particles (roughly 85%) represent assemblies with a single helical turn containing 13 subunits, there is a smaller proportion of particles containing larger number of subunits with up to ~2.5 helical turns. Further classification of particles revealed notable variation in the helical pitch, ranging from 53 Å to 55 Å and ~13.4 subunits per turn of the helix (S3 Fig). For comparison, the Measles nucleocapsid protein, the closest homologue with an available structure, forms helical assemblies with a 50–66 Å pitch and 12.8–13.5 subunits per turn[56]. Such flexibility in the nucleocapsid assembly may facilitate conformational rearrangements required for RdRp access to the nucleocapsid bound RNA during mRNA transcription and genome replication, and/or for binding of host proteins that regulate this process.

Besides the typical spiral assembly, particles that resemble a clam-shaped assembly are also present, with a calculated pitch of ~45 Å with ~13.1 subunits per turn and at least two distinct conformations observed (S7 Fig). Since the clam-shaped assembly has not yet been observed during replication *in vivo* for NiV, it is also possible that the formation of these assemblies were induced by the conditions used during production of recombinant N protein. However, the consistent presence of such assemblies during purification of the NiV nucleocapsid (S8 Fig), indicates an ordered, stable complex, suggesting that they may indeed serve some, as yet unidentified, biological purpose. Indeed, the N protein is known to bind to a short leader transcript and thought to form short ring-like assemblies during the very early stage of viral mRNA transcription and protein translation [57,58]. Two of these short ring-like assemblies may assemble in a face-to-face manner to form a highly stable complex. It is also possible that the interactions underpinning the clam-shaped assembly serve to protect the 5' end of the genomic nucleocapsid through interaction with an N-bound-leader RNA ring, as such assemblies have been purified from isolated MeV virions[58]. Similarly, these assemblies may also lead to the budding of virions which contain multiple RdRp-nucleocapsid assemblies[59]. Similar clam-shaped assemblies were observed for NDV and most recently for the Sendai virus (SeV) (S9 Fig), both for recombinantly produced protein and for nucleocapsids purified from virions, and it has been proposed they may act as a seed for formation of a double headed spiral assembly[14,42], although the possibility cannot be excluded that these NDV and SeV assemblies may have formed from damaged fragments of virionic nucleocapsids[60]. Nevertheless, as NiV N shares only 28% and 29% sequence identity with the NDV and SeV nucleocapsids,

respectively, the observation of similar clam-shaped assemblies indicates their potential biological significance, necessitating further research.

During the paramyxoviral infection, the level of N protein expression is the highest among the viral proteins[61]. As such, N protein displays strong immunogenicity and thus serves as a valuable antigen for diagnostics, serological surveillance and, potentially, vaccine development [62]. Early studies on paramyxoviral N protein had reported that the flexible C-terminus region was susceptible to proteolysis and hence expected to be surface-exposed[63]. As seen from the structure presented here, the last defined residue, E398, although situated at the inner core, points towards the outer edge of the assembly (S6 Fig), indicating that the C-terminal region, residues 399–532, is indeed at least partially exposed within the nucleocapsid assembly. Identification of a handful of antibodies that bound to segments within this disordered C-terminus in recent studies, confirmed the importance of this region as a valuable epitope[64]. In general, however, interactions mediated by disordered protein regions are of lower affinity than with folded areas owing to the thermodynamic costs associated with folding and binding [65]. In this regard, the three-dimensional structure presented here allows identification of areas that are surface exposed, i.e. not buried in protein-RNA or subunit-subunit interactions, and at the same time have a well-defined fold. Moreover, comparison of structural differences and mapping sequence variation between paramyxoviruses to the structure (Fig 4), allows the identification of folded segments that could be potent for specific recognition and ultimately for virus diagnostics. One such region, residue segment 108–142, and specifically helix H5, residues 123–139 (Fig 4), forms a surface exposed area around the outside of the nucleoprotein assembly, suggesting it may serve as a valuable epitope for serological surveillance. In accordance, equivalent segments have been identified as antigenic sites in nucleocapsid proteins of measles virus (residues 122–158 corresponding to 120–156 in NiV N) [66], rinderpest virus (residues 115–150 corresponding to 113–148 in NiV N)[67] and Sendai virus (residues 119–134 corresponding to 113–128 in NiV N)[68].

Comparison with the structure of an RNA-free monomer in complex with a segment of the P-protein[12] indicates a mechanism by which the P-protein can modulate assembly of the nucleocapsid during viral replication. The P-protein segment is bound in a groove which is occupied by the N-terminal helix of one adjacent subunit and a C-terminal region of another adjacent subunit. Thus, binding of the P-protein would directly compete with the nucleocapsid assembly formation, as has been proposed earlier on the basis of the nucleocapsid-RNA structure of the measles virus[12].

The first high-resolution structure of the Nipah nucleocapsid assembly reported here, determined in complex with RNA, will inform the design of inhibitors that disrupt subunit-subunit or protein-RNA interactions. Future studies on N protein interaction with other factors such as the P [61,69] and M [70] proteins will allow understanding of the full scale of molecular events that occur during nucleocapsid assembly and viral replication.

## Supporting information

**S1 Fig. CryoEM data processing workflow.** After the 2D classification, 189,662 particles were initially sorted by 3D classification using an initial model generated from EMDB-0141 low-pass filtered to 60 Å[22]. The best 3D class (squared on the figure) was used as a reference for a new round of 3D classification, to sort the 189,662 particles into the spiral assembly (65%) and clam-shaped assembly (35%) groups. An additional 2D classification was performed to inspect the selected particles and 2D class averages for each type of assembly. For the spiral assembly, a mask representing a single turn of spiral assembly was applied to the map, leading to 3.5 Å structure. Further 3D classification without alignment resulted in several spiral assembly

maps, with different conformations and compositions (S3B Fig). For the clam-shaped assembly, further 3D classification resulted in semi-spiral clam shaped assembly and in a spiral clam shaped assembly.
(TIF)

**S2 Fig. 3D local symmetry refinement of the Nipah N protein spiral assembly.** (A) Local symmetry refinement workflow. (B) "Gold-standard" FSC plot before and after local symmetry refinement. (C) Angular distribution plot for the spiral assembly.
(TIF)

**S3 Fig. Models of the helical and clam shaped assemblies of the NiV N protein.** (A) Ribbon model in white is a duplicate of the helical assembly, generated from the original structure, aligned using Chimera MatchMaker feature, so that its first subunit matches the last subunit of the original structure. The resulting spiral turn was used for calculation of the pitch and the number of subunits per turn. (B) CryoEM maps of the top four 3D classes from the classification of NiV N protein helical assembly (S1 Fig), shown along with ribbon diagrams of fitted N protein subunits. Number of particles contributing to each respective class is indicated above each model. Overlay of all the fitted NiV N protein models reveals a subtle variation in the seam region of the helical turn. (C) Overlay of single subunits from the four different assemblies shown in (B), calculated and shown for two single subunits of each subunit, taken from two different positions of the helical assembly. Scale bar, 1 Å.
(TIF)

**S4 Fig. Comparison of the Nipah N protein structure in the RNA-free and RNA-bound states.** Superimposed models are presented as cartoons. The RNA-free N protein (pdb:4co6) [15] is in light purple; while the RNA-bound protein is in light grey.
(TIF)

**S5 Fig. Multiple sequence alignment of Nipah virus, Hendra virus, Cedar virus and Measles virus.** Multiple alignment graphic was prepared using ESPript 3.0 (http://espript.ibcp.fr/ESPript/cgi-bin/ESPript.cgi).
(TIF)

**S6 Fig. Sequence differences between Nipah virus (NiV), Hendra virus (HeV), Cedar virus (CeV) and Measles virus (MeV) mapped onto the Nipah N protein assembly surface.** Amino acids differing between the NiV and HeV N proteins, are labelled.
(TIF)

**S7 Fig. CryoEM maps for the two major types of clam-shaped assemblies.** (A) Local resolution of CryoEM maps for each of the two assemblies are shown in three different views. Cartoon illustrations for each view are also presented. (B) "Gold-standard" FSC plots for the semi-spiral clam and spiral clam-shaped assembly.
(TIF)

**S8 Fig. Purification of the Nipah N protein.** (A) SEC purification and SDS PAGE profile of Nipah N protein. Elution retention volume of thyroglobulin is indicated. (B) Negative stained EM micrograph of the SEC purified N protein from (A). Dashed red boxes represents side-views of the clam-shaped assembly.
(TIF)

**S9 Fig. Comparison of clam-shaped assemblies formed by the Nipah, Newcastle disease [14], and Sendai viruses[42].**
(TIF)

**S10 Fig. Electrostatic surface potential of the Nipah N protein assembly.** Calculations were performed at pH 7.0 and 150 mM salt concentration. Positive and negative charges are colored in blue and red, respectively.
(TIF)

**S1 Table. Interface area between adjacent protomers of the Nipah virus (NiV), Parainfluenza virus 5 (PIV5) and Measles virus (MeV).** The buried surface area was calculated for the helical assembly of each virus.
(DOCX)

**S2 Table. Domain movements in the NiV N protein around the hinge area associated with RNA binding.** Rotational and translational values were derived from comparison of the RNA-free (pdb:4co6)[15] and RNA-bound states. All values were estimated as described in **Material and Methods**.
(DOCX)

**S3 Table. Sequence identity matrix of Paramyxoviral N protein.** The genus for each virus is indicated in brackets.
(DOCX)

**S4 Table. Statistics of CryoEM data collection, processing and structure refinement**
(DOCX)

**S5 Table. RMSD between the RNA-bound (this study) and RNA-free (pdb:4co6)[15] NiV N protein calculated for Cα atoms.**
(DOCX)

## Acknowledgments

We thank Wen Siang Tan (Universiti Putra Malaysia) for kindly providing the NiV Nucleocapsid gene. We thank Maria Chechik, Sam Hart (University of York), and Svetomir Tzokov (University of Sheffield) for help in CryoEM grid screening. We acknowledge Diamond Light Source for access to- and support from- the CryoEM facilities at the UK national electron bio-imaging centre (eBIC). We are also grateful for computational support received from the University of York High Performance Computing service (Viking) and the Research Computing team.

## Author Contributions

**Conceptualization:** De-Sheng Ker, Alfred A. Antson.

**Data curation:** De-Sheng Ker, Huw T. Jenkins, Sandra J. Greive.

**Funding acquisition:** Alfred A. Antson.

**Investigation:** De-Sheng Ker, Huw T. Jenkins.

**Methodology:** De-Sheng Ker, Alfred A. Antson.

**Project administration:** Sandra J. Greive, Alfred A. Antson.

**Supervision:** Sandra J. Greive, Alfred A. Antson.

**Validation:** De-Sheng Ker, Huw T. Jenkins, Alfred A. Antson.

**Visualization:** De-Sheng Ker.

Writing – **original draft:** De-Sheng Ker.

Writing – **review & editing:** De-Sheng Ker, Huw T. Jenkins, Sandra J. Greive, Alfred A. Antson.

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
