## [Decision Letter · Decision Letter 0]

13 May 2021

Dear Prof Antson,

Thank you very much for submitting your manuscript "CryoEM structure of the Nipah virus nucleocapsid assembly" for consideration at PLOS Pathogens. As with all papers reviewed by the journal, your manuscript was reviewed by members of the editorial board and by  independent reviewers. We are happy to say that the reviewers were enthusiastic about the paper, and provided suggestions for improvement. We are therefore likely to accept your manuscript for publication, provided that you modify it according to the review recommendations.

Sincerely,

Félix A. Rey

Associate Editor

PLOS Pathogens

Christopher Basler

Section Editor

PLOS Pathogens

Kasturi Haldar

Editor-in-Chief

PLOS Pathogens

orcid.org/0000-0001-5065-158X

Michael Malim

Editor-in-Chief

PLOS Pathogens

orcid.org/0000-0002-7699-2064

Reviewer Comments (if any, and for reference):

Reviewer's Responses to Questions

**Part I - Summary**

Reviewer #1: In their manuscript, Ker and colleagues describe three cryo-EM structures of recombinantly produced Nipah virus nucleoprotein associated with RNA and forming distinct assemblies.

The first is a helicoidal one made of 13 sububunit solved at 3.5 Å resolution. As for measles virus, RNA is wrapped around the nucleocapsid protein assembly with a periodicity of six nucleotides per protomer, in the “3-bases-in, 3-bases-out” conformation.

Aside from this helical structure, the authors describe two clam-shaped assemblies of which the resolution is lower (4.3 and 5.2 Å).

These structures, obtained for a very pathogenic virus, complement data obtained with other paramyxoviruses. They confirm what has already been observed with measles.

Furthermore, they reveal how the N- and C-terminal segments of the NiV N protein, which were not present in the previous construct used to determine the RNA-free monomeric NiV N protein (associated with P), stabilize the assembly, by interacting with the same surfaces of adjacent subunits.

This is a nice structural work bringing new information on an important human pathogen a high potential for emergence. The manuscript is easy to read and deserves to be published.

Reviewer #2: In this manuscript, the authors present a 3.5 A resolution cryo-EM structure of a helical turn of the Nipah virus nucleocapsid obtained by heterologous expression of Nipah N in E. coli. Two lower resolution structures of clam-shaped assemblies are also obtained. The structures are compared with other available structures of nucleoproteins of non-segmented negative strand RNA viruses, confirming their similarity. Overall, the manuscript is well written, the technical aspects of the work are well done, and the discussion is insightful.  I only have a couple of minor comments.

**Part II – Major Issues: Key Experiments Required for Acceptance**

Reviewer #1: No major issue

Reviewer #2: None

**Part III – Minor Issues: Editorial and Data Presentation Modifications**

Reviewer #1: I have only some minor comments that may be used by the autors to improve the discussion.

In the discussion, the authors wrote: “Most strikingly, some viruses, like Rabies, a member of the Rhabdoviridae, bind the RNA inside the helicoidal assembly (45) in contrast to the viruses of the Paramyxoviridae (e.g. NiV), Filoviridae(42, 46) and Pneumoviridae(47, 48) families, where the RNA is wrapped around the outside."

First, reference 45 is related to VSV N. The correct reference to rabies virus N is Albertini et al. (Science. 2006 Jul 21;313(5785):360-3.) which has to be quoted too.

However, those two structures of rhabdovirus nucleoproteins are obtained from crystal rings. In the virions, in the RNP assembly, the proteins are tilted (compared to their organization in the rings) and thus the RNA is not really positioned inside the helicoidal assembly. See Ge et al. (Science. 2010 Feb 5;327(5966):689-93) for VSV and Riedel et al. (Sci Rep. 2019 Jul 3;9(1):9639) for rabies virus.

Interestingly, reminiscent of the organization of clam-like assemblies of NIpah N, in the asymmetric unit or rabies virus nucleoprotein crystal structure (PDB code: 2GTT), there are two rings, made of 11 nucleoproteins, that are associated head to head through contacts involving their N-terminal lobes.

Second, when comparing rhabdovirus and paramyxovirus, it might be pertinent to discuss the N0P structures. For paramyxovirus, as mentioned by the authors, the P-protein segment is bound in a groove which is occupied by the N-terminal helix of one adjacent subunit and a C-terminal region of another adjacent subunit. However, for rhabdoviruses, the N-terminal part of P occupies the RNA binding site (Leyrat et al. PLoS Pathog. 2011 Sep;7(9):e1002248.). Therefore, the way the N is kept soluble and RNA-free by P is slightly different for paramyxoviruses and rhabdoviruses.

Reviewer #2: 1) While describing the symmetry of the nucleocapsid helices, the authors call it helicoid, helicoidal or helical. I would suggest to stick to the conventional term, helical, throughout the manuscript.

2) In the second paragraph of the introduction, N is introduced as nucleocapsid (N) protein. It should be nucleoprotein.

3) I would suggest to change the word order in the following sentence (page 2) to make it clearer: ‘The only available structural information for the NiV nucleocapsid is for a truncated single subunit lacking N-terminal (residues 1-31) and C-terminal (residues 384-532) regions, which was determined in complex with a 50 amino acid peptide of the P protein(15).’ => ‘The only structural information for the NiV nucleocapsid is available for a truncated single subunit lacking N-terminal (residues 1-31) and C-terminal (residues 384-532) regions, which was determined in complex with a 50 amino acid peptide of the P protein(15).’

4) Would it be possible to change the color of the C-terminal arm so that the reader can more easily see it in the figures? In the present version, the colors or the C-terminal domain of the Ncore and of the C-terminal arm are not easy to distinguish, and in particular in the Figure 1 one would like to be able to clearly see the position of the C-terminal arm that, in the nucleocapsid helix, should be situated between two successive helical turns.

5) The current discussion about the possible relevance of the clam-shaped assembly is very interesting. In addition, a very recent paper (Zhang et al., Communications Bioloby 2021) describes the structure of the double-headed nucleocapsids from Sendai virus and presents a map of the clam-shaped structure which is better resolved than the one of the Nipah virus. It seems important to compare the two structures (the map is deposited in the EMDB) and to enrich the discussion by including the new information offered in the Sendai virus work.

6) Please check the legend of the Figure S3A and state what exactly is on the left and on the right.

PLOS authors have the option to publish the peer review history of their article (what does this mean?). If published, this will include your full peer review and any attached files.

Reviewer #1: No

Reviewer #2: No

Figure Files:

Data Requirements:

Reproducibility:

References:

---

## [Editor Report · Decision Letter 1]

22 Jun 2021

Dear Prof Antson,

We are pleased to inform you that your manuscript 'CryoEM structure of the Nipah virus nucleocapsid assembly' has been provisionally accepted for publication in PLOS Pathogens.

Best regards,

Félix A. Rey

Associate Editor

PLOS Pathogens

Christopher Basler

Section Editor

PLOS Pathogens

Kasturi Haldar

Editor-in-Chief

PLOS Pathogens

orcid.org/0000-0001-5065-158X

Michael Malim

Editor-in-Chief

PLOS Pathogens

orcid.org/0000-0002-7699-2064

The original version of this manuscript was already very interesting, and the reviewers had only minor suggestions. The current version is improved and clearer, I have no additional comments to make.
---

## [Editor Report · Acceptance letter]

12 Jul 2021

Dear Prof Antson,

We are delighted to inform you that your manuscript, "CryoEM structure of the Nipah virus nucleocapsid assembly," has been formally accepted for publication in PLOS Pathogens.

Best regards,

Kasturi Haldar

Editor-in-Chief

PLOS Pathogens

orcid.org/0000-0001-5065-158X

Michael Malim

Editor-in-Chief

PLOS Pathogens

orcid.org/0000-0002-7699-2064